# STRIDED TRANSFORMERS FOR PARTIALLY-PARALLELIZED INFERENCE

## ABSTRACT

Auto-regressive large language models have dramatically improved performance in natural language generation tasks. Popular architectures such as the Transformer have enabled parallel training across tokens and scaled to large corpora of datasets. Generation–however–remains a fundamentally serial task where a token must be fully predicted before processing of the next token begins. In this work, we propose a framework for partially-parallelized large model inference by striding autoregressive dependencies between model layers, yielding strategies to improve latency in either memory or compute bound workflows, while preserving fully parallel training. The associated models require a simple modification in training by rolling representations along the sequence axes and create a favorable setup in inference with only minor degredation in accuracy.

## 1 INTRODUCTION

Scaling large language models has enabled remarkable performance on a range of natural language generation tasks, with models such as Palm V2 (Anil et al., 2023) and GPT4 (OpenAI, 2023) showcasing wide-scale utility in a variety of applications (Singhal et al., 2023). These models are largely based on the popular Transformer architecture (Vaswani et al., 2017). By the design of self-attention, training is highly parallel and performed concurrently over all input tokens within a sequence, and recent work has scaled training to hundreds of devices at a time (You et al., 2019). Decoding–on the other hand–is limited by its fundamentally serial nature. Applications of large language models have required careful optimizations to deal with downstream latency constraints (Thoppilan et al., 2022)–for example from conversational agents–or to yield high-throughput batched computations in the case of offline inference (Pope et al., 2023). This paper challenges the assumption that tokens need to be processed one-at-a-time and showcases how minor adjustments to architectures can enable partially-parallelized inference.

With discovered trends in scaling laws (Kaplan et al., 2020), models have dramatically increased in size over previous years, to the point where loading model parameters and transferring these weights to the compute cores can represent a significant laggard in processing pipelines. This is particularly a problem for single example inference (e.g. for on-demand inference) which requires the entire model to be loaded for every token. On the other hand, for sufficiently optimized memory layouts and large enough batches, compute requirements for the network from the self-attention and large multi-layer perceptrons represent the substantial fraction of inference workflows. These two scenarios define what we will refer to as either memory-bound or compute-bound workflows, where any modifications targeting improved latency must operate.

In this paper, we set out to enable partially-parallel decoding by modifying the dependencies required during training, in an effort to improve Transformer-based inference costs. Causal architectures in their simplest form across all layers of a model require the current token to begin the next step of decoding. What if, instead, processing begins with further context (e.g. greater than 1 token away) and does not wait on the most recent token? In such a model, a set fraction of the capacity could be used to process further context and create contextualized representations; later, the remaining model capacity could be used to finalize next-token predictions. Such a framework enables partially-parallel decoding as tokens can start being processed before the prior token has been fully determined. We will refer to this setup as *strided* decoding.

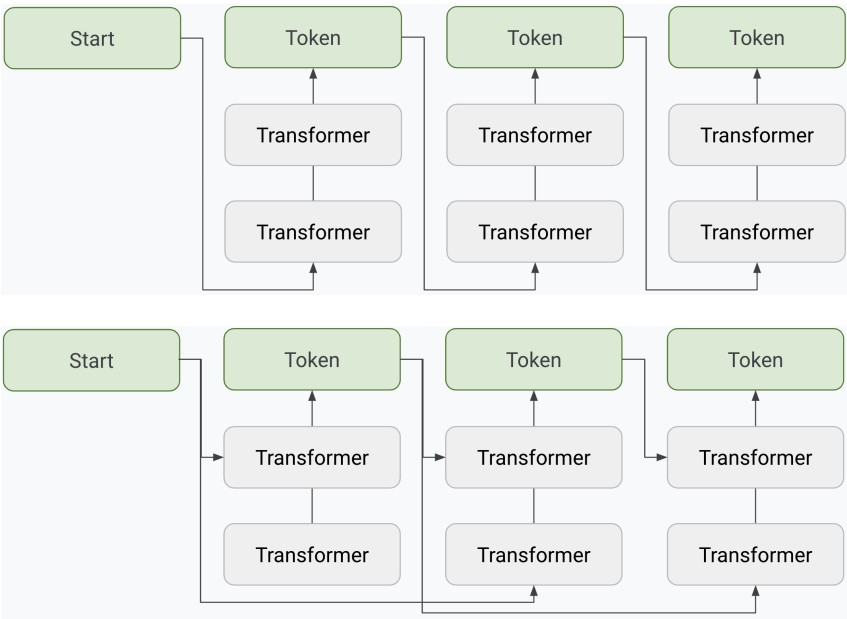

Figure 1: Schematic diagrams of the token interactions of (top) standard Transformer inference and (bottom) strided inference. Note, we do not show the reliance on past history via-self attention. The dependency structure of the proposed architectural modification enables partial processing of tokens before the prior is fully-complete, enabling parallelization.

This strided framework motivated by parallelizing stages of decoding requires a simple yet significant modification to the training setup. Representations need to be 'rolled' along the sequence length after a user-defined number of layers that defines the partial limits of parallelization. To realize the performance of this new strategy, we construct strategies for addressing both memory-or-compute bounded workflows. Memory-bound workflows utilizing strided layouts achieve higher throughput via re-utilization of model weights and rolling out partial processing of multiple tokens. In compute limited-workflows, strided decoding enables an additional axis of parallelization, allowing for lower latency via a greater number of devices.

Our contributions can be summarized as follows:

1. **Strided framework:** In Section 2, we propose the strided framework and detail the adaptations needed for causal dependencies to match during training and inference. Training maintains the *fast* parallelization enabled via the Transformer architecture and only requires a simple *roll* of representations at specified layers during training.

2. **Memory-Bound Inference:** We propose strategies for partially-parallelizing inference in memory-bound workflows by allowing partial processing of multiple tokens at a time (e.g. perform layer 0 for multiple tokens). This strategy is referred to as horizontal parallelization and applies to scenarios where costs are dominated by loading model parameters.

3. **Compute-Bound Inference:** For compute-bound workflows, strided decoders enables what we refer to as vertical parallelization. This strategy is apt for scaling to multiple devices, suggesting a method to reduce latencies in batched inference scenarios where computation is the dominant factor of performance.

## 2 STRIDED TRANSFORMER

We start by considering a standard objective of large language models: next-token prediction. The common assumption presumes the current token interacts with prior context, transforming word embeddings into ones that interact with the history; the contextualized representation is then used

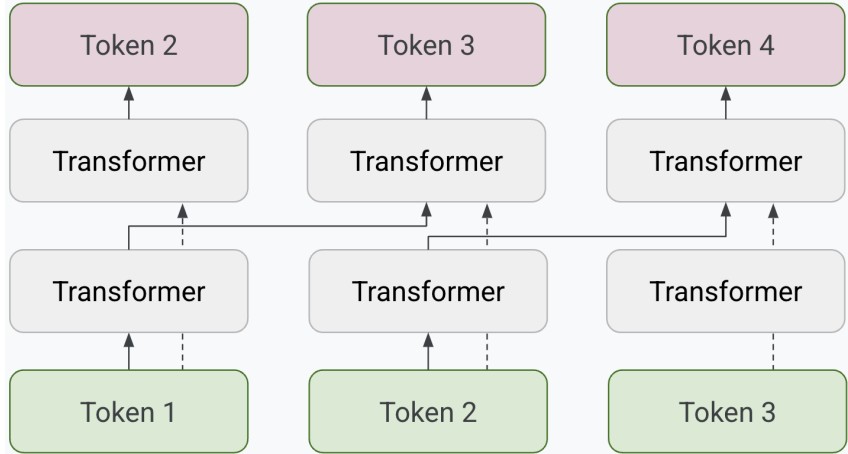

Figure 2: Schematic diagrams of training for strided Transformer models. Note the dependency structure utilized for parallel inference is enabled via a 'roll' along the axis of the sequence length. Dotted-arrows correspond to introduced weighted residual connections that serve to introduce token embeddings after shifts.

to predict the next-token. How consequential–however–is the most recent token? Could we instead process the elongated context first?

This framing motivates the strided transformer, where representations of a token $i$ begin with the embedding of tokens $i - k$ (for a chosen value of $k$). Progressively, over the depth of the associated model, the embedding of the remaining $k$ tokens can be introduced. If the standard formulation of auto-regressive modelling writes next token prediction as $\hat{x}_i = f(x_{<i})$, then the strided framework when $k = 2$ introduced above splits model capacity so that it can be written as $\hat{x}_i = g(x_{i-1}, h(x_{<i-1}))$. This interaction adapts the amount of computation that must wait on $x_{i-1}$, as the model is now split into two components that can be parallelized with the completion of each token (in other words, the computation of $h$ can start before $x_{i-1}$ is decided). Moreover, we could further separate the function $h$ to further improve the degree of parallelization. With this motivation, we describe the modifications needed for training and inference that formalize these ideas.

## 2.1 ROLLING ALONG AXES

For training, we propose a simple modification that introduces the dependency structure of strided transformers, while maintaining the efficiency of parallel training of large language models. For simplicity, we will first consider a 2-layer Transformer and a stride of length 2 so the first layer only depends on $x_{<i-1}$ and the second layer integrates the representation of $x_{i-1}$, as defined above. To enable this dependency structure in training, we simply need to roll representations along the sequence axis after the first layer and can otherwise maintain the rest of the Transformer architecture. In this contrived example, the first layer can be thought of as $h$ and the roll allows for the interaction of $h(x_{<i-1})$ and $x_{i-1}$ to create desired dependencies. A visualization of the training procedure is provided in Figure 2.

Notably, this framework for striding dependencies is more general than the 2 layer model discussed above. First, we can choose to perform rolls after arbitrarily many Transformer layers, and moreover, we can also choose to shift by any desired sequence length. As we will see in the associated decoding frameworks in Section 3, the design decisions around strides depend heavily on the expected usage of a given model and the desired trade-offs between accuracy and latency, as we explore in the experiments section of this paper. Nevertheless, these models maintain the framework that early layers of a Transformer can process further context before the more recent tokens become available.

## 2.2 TOKEN MIXING AFTER STRIDES

Standard practice in Transformer architectures updates the original embedding with a residual layer after each attention and MLP operation. In the strided formulation, we however introduce a discontinuous change as a result of the roll along the sequence length and must design a strategy to integrate the rolled embedding that arises from processing elongated context with the new token embedding made at this later layer of the model. For example, in Figure 1, the second layer must integrate the rolled representation of $h(x_{<i-1})$ and $x_{i-1}$ before creating predictions. We found that a simple Layernorm to perform well as this task. Inputs after strides take the form:

$$\text{Layernorm}((1 - \lambda)h(x_{<i-k}) + \lambda x_{i-k}))$$

where $k$ is a variable used to denote stride length and does not need to equal 1 as in prior examples. This design enables the model to balance between processed context and the current token when proceeding with model inference. Notably, $\lambda = 1$ would correspond to entirely forgetting context and only model capacity after the last roll would be utilized.

## 3 DECODING

Decoding remains a bottleneck of large-scale transformer inference due to its serial nature; a major goal in striding Transformer tokens is to enable faster inference in these types of large-scale Transformer networks by enabling parallelization from the limited-dependency structure. We will–however–have to separately consider the decoding scenarios in memory-bound vs. compute-bound workflows due to the distinct requirements of each.

### 3.1 MEMORY-BOUND

Small-batch inference of large language models requires significant memory bandwidth due to the need to load model weights into compute cores; examples such as on-demand inference for dialog applications leads to tight constraints on the latencies that models can have. Fixing hardware, this typically leads to lower sequence lengths from inference or smaller models being utilized in order to meet the desired constraints. By striding the dependency structure of large Transformer architectures, we showcase how multiple tokens can be partially processed at once. Due to the horizontal interaction depicted in Figure 3, we refer to this strategy as horiztonal parallelization.

We again start with the simplified setup of a 2-layer network that is depicted in Figure 2. In order to limit the number of times model weights need to be passed to compute cores, we argue that the first layer (with a stride of 2) can be parallelized so that two tokens are processed at a time, as the later does not need the previous token for this stage of processing. Notably, the KV-caches of the corresponding layer are also shared, reducing any additional costs associated with this component of decoding. The processing of these two tokens can therefore be effectively vectorized up to the correction required for causal-attention on the second token. We then note that the second layer of the network must still be performed serially due to the dependence on previous token as input. In this scenario, we see a theoretical 25% improvement in inference speed as about half the model only needs to be loaded per every other token. We visualize this algorithm in Figure 3.

Notably, horizontal parallelization is quite flexible. As the focus is on loading as few of the model weights as possible, layers can be considered independently for the degree of parallelization. For example, the first layers could choose a stride of 8 for 8-fold parallelization; the only requirement is that the last layer have a stride of 1 to allow for the entire history to be included before next token prediction. Clearly, this allows for significant flexibility into choosing the desired accuracy vs. degree of parallelization tradeoffs that are applicable in any given scenario. To push past the 25% inference speedup in the simplified example, we consider to further reduce memory loads, at the potential expense of accuracy. These extensions are discussed and explored in Experimental Results.

### 3.2 COMPUTE-BOUND

Next, we note that strided transformers also allow for a form of pipeline parallelism. Figure 4 showcases a simple pattern that we refer to as *vertical* parallelization where all relevant partial processing

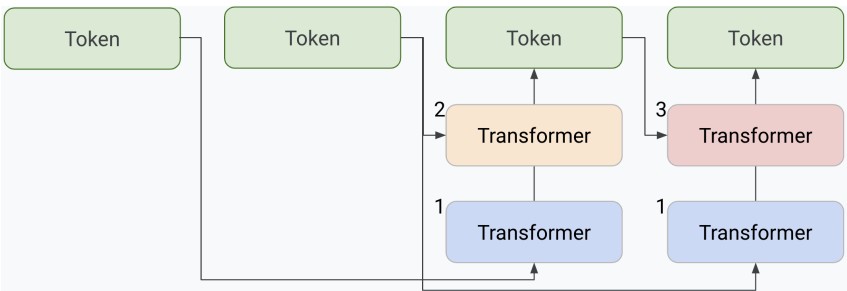

Figure 3: Schematic diagrams of horizontal parallelization allowed by strided decoding, enabling fewer memory loads in the strided layers of the Transformer models. Numbers and colors correspond to parts of the model that can be performed in parallel. Where the standard Transformer model would need to load the weights of 2 distinct layers per token, the strided decoder is able to share the weights for first layer and generate activations 2-at-a-time (1, blue). The second layer however must be performed serially as the token prediction as a result of (2, yellow) is require as input to (3, red). This corresponds to model weights loaded 3 times, a total improvement of 25%.

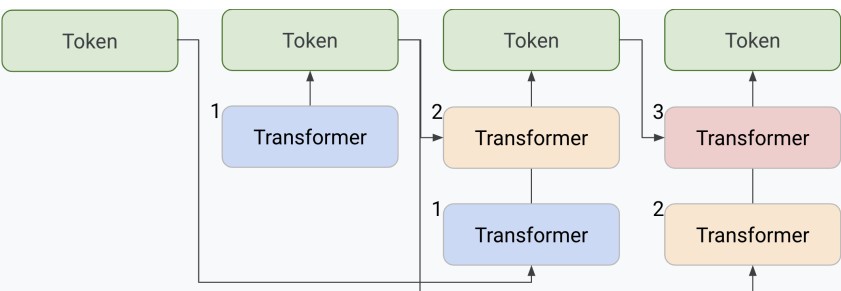

Figure 4: Schematic diagrams of vertical parallelization allowed by striding Transformer inference. Numbers and colors correspond to parts of the model that can be performed in parallel. In this form of parallelization, components organized diagonally can be run in parallel. Vectorization of the associated computations could be useful if the devices could support additional computation (e.g. insufficient flop utilization); otherwise, strided transformers enable another form of device parallelization, providing an additional axes to parallelize Transformer inference.

| Layers | Maximum Stride | Maximal Memory Speedup | Maximal Compute Speedup | Test Loss |
|---|---|---|---|---|
| 3 | 1 | 50% | 50% | 0.740 |
| 6 | 1 | 0% | 0% | 0.681 |
| 6 | 2 | 25% | 50% | 0.701 |
| 6 | 1 | 50% | 50% | 0.696 |
| 12 | 1 | 0% | 0% | 0.640 |
| 12 | 2 | 25% | 50% | 0.655 |
| 12 | 1 | 50% | 50% | 0.640 |
| 24 | 1 | 0% | 0% | 0.619 |
| 24 | 2 | 25% | 50% | 0.612 |
| 24 | 1 | 0% | 0% | 0.605 |
| 24 | 2 | 25% | 50% | 0.616 |

Table 1: Evaluation of the strided Transformer on a deconder-only benchmark based on a WMT EN-GE benchmark. The strided Transformer architecture showcases a favorable tradeoff in performance for speedups, compared to baselines such as decreasing model size.

begins as soon as a token is predicted. In the consistent example, $h(x_{<i})$ and $g(x_{i-1}, h(x_{<i-1}))$ can both begin as soon as token $i - 1$ is predicted, and if strides in the original model were chosen to be equally spaced, then this corresponding operation can be appropriately parallelized. In particular, the promising avenue for this form of parallelizaiton is when using multiple devices. Notably, we do not think that this form of parallelism supersedes model parallelism as it does require a trade-off in accuracy for faster performance. However, the two could be used in extreme cases for faster inference overall.

## 4 EXPERIMENTAL RESULTS

Strided transformers reduce dependencies in early layers of Transformer models, enabling parallel processing of multiple tokens at inference. The designed setup is expected to reduce model performance due to reduced capacity placed on the most recent tokens; in this section–however– we showcase the latency tradeoffs may be favorable when compared to alternatives such as reducing model depth. Table 1 presents the test performance for models trained on the wmt_t2t_ende_v003 task from the T5x paper (Roberts et al., 2022) and the corresponding theoretical inference speedups achievable via improved parallelization. For a first evaluation, we present results where a single-stride is placed halfway into the model (corresponding well to the 2-layer model discussed throughout the paper). We note that minimal degradation in performance is seen for 25% latency improvements in memory-bound workflows, paralleism across multiple devices would also lead to a 2x improvement in latency (disregarding communications costs) with comparable tradeoffs.

### 4.1 BEYOND SIMPLE STRIDES

Initial results tested on small strides already showcase favorable parallelization properties when compared to alternatives such as reducing depth. Nevertheless, it still may be of interest to include more than a single stride, halfway into the model: either by increasing the number of layers that are parallelized or repeating strides within a single model. We explore the axes and the effect of parallelization vs. performance in the rest of this section.

**Minimally Serial**  Horizontal inference arising from memory-bound workflows do not set a limit on the number of layers that should consider every token (of stride 1). In early experiments, we divided the model in half and placed a single stride operation to assess performance, yielding an improvement of 25% in parameter loading. Now, however, we can adjust the stride to shift later in the model, tuning the model towards context and providing less-and-less capacity to process the current token. In Figure 5b, we showcase the results of moving the stride operation from layer 1 to layer 11 so that more-and-more of these layers could produce tokens 2-at-a-time. We show minimal degradation in performance arises even as a stride arise deeper within the model.

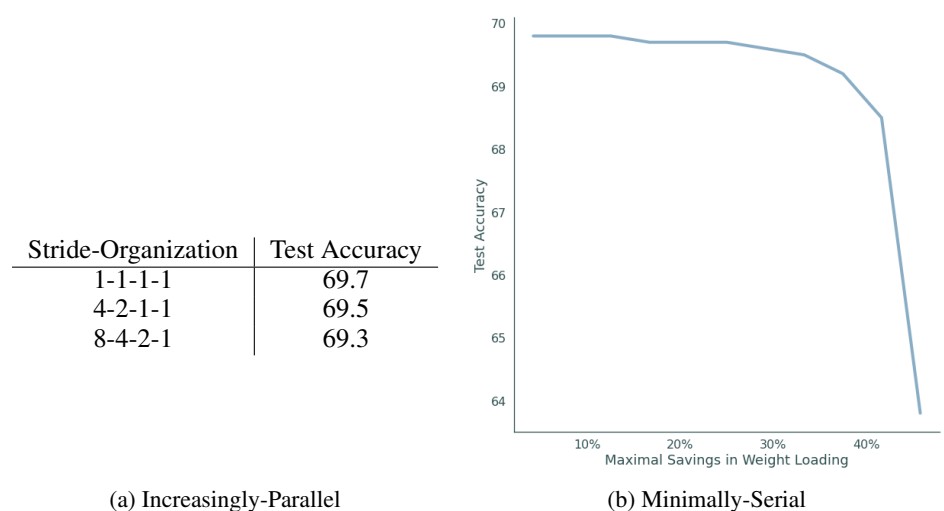

| Stride-Organization | Test Accuracy |
|:---:|:---:|
| 1-1-1-1 | 69.7 |
| 4-2-1-1 | 69.5 |
| 8-4-2-1 | 69.3 |

(a) Increasingly-Parallel                    (b) Minimally-Serial

Figure 5: Evaluation of the strategies for increasing parallelization in memory-bound scenarios: (a) first exploring increasingly parallel adapations and (b) then designing strides so that as few model parameters are serial.

| Layers | Maximum Stride | Memory Speedup (Decoder-Only) | Compute Speedup (Decoder-Only) | Test Accuracy |
|:---:|:---:|:---:|:---:|:---:|
| 24 | 1 | 0% | 0% | 66.6 |
| 24 | 2 | 25% | 50% | 65.8 |
| 24 | 3 | 38% | 66% | 64.6 |
| 24 | 4 | 48% | 75% | 64.1 |

Table 2: Evaluation of the strided Transformer on a encoder-decoder benchmark.

**Increasingly Parallel**  For horizontal parallelization in memory-bound workflows, it is also possible to choose arbitrary stride lengths. Resulting strides of length $k$ suggest that $k$-tokens can be processed at once, and so going beyond 2 promises an avenue to surpass 25% improvement even if the stride is only applied to half the model. For this experiment, we divide a model into 4 equal subcomponents and study the test accuracy tradeoff improving form the standard 25% latency improvement in the rest of the model. In Figure 5a, we see that two strategies of increasing the parallelization do not drastically harm test accuracy. In particular, the stride organization of 8-4-2-1 is of interest as this pushes the theoretical improvements in memory bound workflows past 53% with minimal degradation in performance.

## 4.2  ENCODER-DECODER MODELS

Finally, we have so-far discussed adjustments to the casual decoder-only architectures. In this section, we note that the same modifications can be made in text-to-text applications with encoder-decoder architectures. For this setup, we used the base examples from the original T5 paper (Roberts et al., 2022) and train text-to-text models on the c4_span_corruption task, where the decoder includes the strided formulization that is presented in the rest of the paper. Notably, in Table 2, we see that the performance displays similar trends to the decoder-only models, with minimal degradation at strides of length 2 in halfway into the model. Greater parallelization can be achieved for greater accuracy tradeoffs.

## 5 RELATED WORK

### 5.1 PARALLELISM

Parallelism for Transformer-based architectures has received widespread attention. Training-based efforts include NeMo (Kuchaiev et al., 2019), Megatron (Shoeybi et al., 2019), GSPMD (Xu et al., 2021), and Alpa (Zheng et al., 2022). Core concepts from these works include tensor parallelism and memory optimizations. GPipe (Huang et al., 2019) also introduced pipeline parallelism for high-throughput training but provides limited differentiation with inference. Other strategies for efficient inference include minimizing padding or packing sequences. Of particular interest is pipeline parallelism which most closely resembles the large-batch, compute-limited evaluations. However, this work is fundamentally different as it does not require the same form of parallelism during training.

### 5.2 LANGUAGE MODEL INFERENCE

Efficient Transformer inference has been a topic of widespread popularity, and a number of efforts have attempted to design more efficient attention layers (for speeding up both training and inference). Alternate approaches include compressing model weights via pruning Liang et al. (2021), quantization Dettmers et al. (2022) or distillation (Hinton et al., 2015). This work is largely orthogonal to these prior efforts, providing another possible avenue for improving parallelization during inference.

## 6 CONCLUSIONS

Scaled large Transformer models promise novel capabilities and tooling via a natural language interface. However, despite the advantages in performance at scale, latency-sensitive use-cases lead to tight constraints that end up leading to performance trade-offs. In this work, we propose an orthogonal method of parallelization to prior work via strided layouts, where more recent information is incoporated deeper within a model rather than at the input layer, yielding the proposed parallelization over depth.

Overall, we present a strategy for improving parallelizing language model inference via modifying causal dependencies at a per-layer basis. We however, note that the resulting architecture maintains approximately the same total FLOPs (total compute cost); alternate strategies such as mixture of experts (Fedus et al., 2022), adaptive computation including early termination (Ainslie et al., 2023), or tied weights (Lan et al., 2020) all are promising avenues for the overall computation required for language models. We believe that these applications can be used in conjunction with strided transformers for low-latency efficient prediction.

### 6.1 REPRODUCIBILITY STATEMENT

The experiments performed in this work correspond to well-tested benchmarks within the machine learning community, and all adjustments to standard models have been described thoroughly in this work.

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
