# OpenReview forum: "Strided Transformers for Partially-Parallelized Inference"
_ICLR.cc/2024/Conference — ICLR 2024 Conference Withdrawn Submission_

### Official Review · Reviewer_YvXk · 2023-10-19

**Soundness:** 2 fair
**Presentation:** 2 fair
**Contribution:** 2 fair
**Rating:** 3
**Confidence:** 4

**Summary:**

This paper introduces a partially-parallelized Transformer model called the Strided Transformer. Unlike traditional autoregressive Transformers, which require a token to be fully predicted before predicting the next token, the Strided Transformer feeds the predicted token simultaneously to several subsequent steps and inserts it into different layers. In this way, strided Transformer supports parrallel computation of different time steps during decoding, which accelerate the decoding and reduces the memory load.

**Strengths:**

1. The proposed approach is innovative and intuitive.
2. Based on the current results, the Strided Transformer achieves a favorable speedup with only a relatively small cost of accuracy. Its performance appears promising.

**Weaknesses:**

1. This work is too hasty and lacks sufficient experiments to support the conclusion. I would raise my score if the authors could provide a fully prepared version during the rebuttal stage.
2. There is a lack of discussion and comparison with related work, such as non-autoregressive [1] or semi-autoregressive [2] sequence generation approaches.
3. The authors report only the theoretical decoding speedup, without evaluating its performance in a real-world environment.
4. Reporting test losses on WMT benchmarks is not a standard practice. Please implement greedy or beam search and provide the corresponding BLEU scores for a more meaningful evaluation.


[1] Gu, Jiatao, et al. "Non-autoregressive neural machine translation." arXiv preprint arXiv:1711.02281 (2017).

[2] Wang, Chunqi, Ji Zhang, and Haiqing Chen. "Semi-autoregressive neural machine translation." arXiv preprint arXiv:1808.08583 (2018).

**Questions:**

Why do some results in Table 1 appear inconsistent? For instance, the Layer 6 Stride 1 setting has a test loss of 0.681 in line 2, but a test loss of 0.696 in line 4.

---

### Official Review · Reviewer_6i1Q · 2023-10-29

**Soundness:** 1 poor
**Presentation:** 2 fair
**Contribution:** 2 fair
**Rating:** 3
**Confidence:** 4

**Summary:**

The authors introduce the Strided Transformer architecture, which reduces token dependencies in the initial layers of the Transformer model. This design allows for partial parallelization during the decoding process and exhibits good theoretical speedups in both memory-bound and compute-bound settings.

**Strengths:**

The proposed Strided Transformer architecture exhibits good theoretical speedup properties.

**Weaknesses:**

The work is incomplete, and the experiment is not sound.
1.  The authors only provide a theoretical evaluation of speedup. It seems they did not implement or perform experiments.
2. Table 1 only presents test loss as a metric for performance degradation, without including any generation results such as BLEU scores to assess the quality of output.
3. The work is not ready for publication, especially given that it consists of only 8 pages with one page containing only two figures.

**Questions:**

The idea is quite fancy. Comprehensive experiments are needed.

---

### Official Review · Reviewer_AsVD · 2023-11-05

**Soundness:** 1 poor
**Presentation:** 1 poor
**Contribution:** 1 poor
**Rating:** 1
**Confidence:** 5

**Summary:**

This paper introduces a partially parallelized inference for Transformer decoders called Strided Transformers. However, the authors completely ignored previous work on (partially) parallel inference with Transformers.

**Strengths:**

1. Parallel decoding with Transformers is an important problem that is worth exploring.

**Weaknesses:**

1. The manuscript completely ignored previous literature on parallel decoding with Transformers, non-autoregressive Transformers, partially parallel decoding with Transformers, and accelerating Transformer decoding.

2. The paper is not very well written. The methodology is not very clearly described and the Figures are not very clear as illustration. Many sentences are not very natural or easily understandable too.

3. The experiments are not convincing because they are not conducted on standard datasets nor compared with meaningful baselines.

**Questions:**

N/A